# Association between Physicians’ Perception of Shared Decision Making with Antibiotic Prescribing Behavior in Primary Care in Hubei, China: A Cross-Sectional Study

**DOI:** 10.3390/antibiotics9120876

**Published:** 2020-12-08

**Authors:** Dan Wang, Chenxi Liu, Xuemei Wang, Xinping Zhang

**Affiliations:** School of Medical Management and Health Management, Tongji Medical College of Huazhong University of Science and Technology, Hangkong Road No. 13, Wuhan 430030, China; 2019511009@hust.edu.cn (D.W.); d201881354@hust.edu.cn (X.W.)

**Keywords:** communication, decision-making, general practice, patient involvement, physician–patient relations, primary care

## Abstract

Shared decision-making (SDM) has been advocated as one effective strategy for improving physician–patient relationships and optimizing clinical outcomes. Our study aimed to measure physicians’ perception of SDM and establish the relationship between physicians’ perception of SDM and prescribing behavior in patients with upper respiratory tract infections. One cross-sectional study was conducted in Hubei Province from December 2019 to January 2020. The SDM questionnaire and prescription data of 2018 from electronic health records data were matched for each physician in this study. Multilevel modeling was applied to explore the relationship between physicians’ perception of SDM and antibiotic prescribing in primary care. Analyses were statistically controlled for demographic characteristics of the physicians and patients. Physicians’ positive perception of SDM had small but statistically significant effects on lower prescribing of antibiotics in the patient group aged over 40 years (odds ratio (OR) < 1; *p* < 0.05). Moreover, female physicians (OR = 0.71; *p* = 0.007) with higher educational levels (bachelor’s degree and above; OR = 0.71; *p* = 0.024) were significantly associated with the prescribing of less antibiotics (*p* < 0.05). A more positive perception of SDM was demonstrated as one significant predictor of less prescribing of antibiotics in the patient group over 40 years. There may be a promising focus of implementing SDM strategies targeting physician–patient communication in primary care.

## 1. Introduction

Increasing antibiotic resistance is recognized as a major threat to global health and is related to antibiotic prescription rates in primary care [1,2]. Acute respiratory infections are one of the most common reasons for consultation in primary care, with upper respiratory tract infections (URTIs) accounting for the largest source of inappropriate antibiotic use [3,4,5,6]. For most patients with URTIs, they still receive antibiotics with limited benefits, as most cases are viral and self-limiting, and antibiotics do not help shorten the illness duration [7]. In China, the setting of this study, there were over 80% of URTI visits to primary care involving an antibiotic prescription, and these prescriptions were often inappropriate [8].

Shared decision-making (SDM) is an approach where a physician and the patient make a decision together on the basis of the best available evidence and considering the patient’s values and preferences [9]. There is increasing evidence recognizing this as one effective strategy for reducing the overuse of antibiotics in primary care [10,11,12,13]. One recent Cochrane review indicated that SDM-promoting interventions significantly reduced antibiotic prescribing for acute respiratory infections in primary care [10].

SDM provides the opportunity for physicians and patients to choose more appropriate treatment options in medical encounters, including the decision to not use antibiotics [7,10]. With full consideration of the trade-off between the benefits and harms of antibiotics, most patients prefer more conservative treatment options [14].

However, SDM does not occur consistently in clinical encounters [15,16,17]. Physicians’ perceptions and attitudes are the key challenges of the implementation of SDM [18]. Many physicians feel they already involved the patient in treatment decisions, or physicians perceived themselves as “decision-makers” for patients [18]. To increase congruence between SDM theory and clinical practice, a thorough understanding of physicians’ perception of SDM is in high demand.

Existing research has revealed that most studies of SDM were conducted in academic centers or public hospitals [19]. A relatively small proportion of studies measured physicians’ perception of SDM in primary care, and most of them involved relatively small groups of samples [20]. The physicians showed a preference towards SDM, although the level of supporting SDM depends on the clinical scenario, patient characteristics, and available treatment options [20].

In China, a few researchers tried to introduce the SDM intervention in a small sample of physicians in teaching hospitals, measuring their preference of SDM—most participants reported a positive preference towards SDM and decision aids [21,22]. However, there is still a lack of understanding as to how physicians perceive decision responsibilities and shared decision-making in medical encounters in primary care.

On the other hand, interventions aiming at promoting SDM have been shown to reduce antibiotic prescription in primary care; however, a limited number of studies have explored the relationship between SDM and fewer antibiotics with a focus on the general judgment of SDM, rather than for specific patients in specific consultations [23]. Whether physicians’ perception of SDM is associated with physicians’ antibiotic prescribing in specific consultations still requires further exploration. 

Therefore, our study aimed (a) to assess physicians’ perception of SDM with patients with URTIs in primary care in China, and (b) to explore the association between physicians’ perception of SDM and antibiotic prescribing in practice. 

## 2. Results

### 2.1. Demographic Characteristics of the Study Sample

In sum, the dataset of 121,481 prescription data files nested in 264 primary care physicians was matched with completed SDM-Q9 measurements for each physician. Among the 264 physicians, 64.4% were male and 35.6% were female. The majority of physicians had vocational or bachelor’s degree (83.0%), while only a small proportion of physicians received secondary vocational education (17.05%). Around two-thirds of the participants were from general practice (67.1%), having an average of 17 years of working practice. The demographic characteristics of primary care physicians are presented in Table 1.

### 2.2. Physicians’ Perception of SDM and Antibiotic Prescribing Behavior

The average overall SDM score was 74.98 (SD = 12.08), and 25th to 75th percentile was 66.7–88.2. Among the nine statements of SDM, “I precisely explain the advantages and disadvantages of the treatment options of my patients” had the highest mean score of 4.15 (SD = 0.71), while the statements of “I want to know exactly how the patients want to be involved in making the decision” (mean = 3.38, SD = 1.21) and “I thoroughly weighted the different treatment option with my patients” (mean = 3.37, SD = 1.21) were rated with relatively low scores among physicians (Table 2).

In sum, the antibiotic prescribing rate was 61.6% for patients with URTIs. Physicians with a higher education prescribed significantly fewer antibiotics (*p* < 0.001), in which physicians with bachelor education and above had antibiotic prescribing rates below 60%. General practitioners had a significantly higher percentage of prescribing of antibiotics than physicians in other specialties (*p* = 0.037). In addition, facility type was also one significant factor of prescribing antibiotics. Physicians in community health centers (CHCs) had a much lower antibiotic prescribing rate than those of township health centers (THCs) (0.52 vs. 0.67). The results of comparing antibiotic prescribing rates among different physician groups are presented in Table 1.

### 2.3. Relationship between Perception of SDM and Antibiotic Prescribing

The multilevel modeling showed that SDM in the reference group (age < 40 years, male) would contribute slightly positive but not significant effects on more antibiotics prescribing (odds ratio (OR) = 1.01, *p* = 0.070). The interactions between physicians’ perception of SDM and age were significant associated with prescribing antibiotics. Specifically, in the patient group over 40 years (40–65 years group: *R* = 0.99, *p* = 0.021; over 65 years group: OR = 0.98, *p* = 0.002), physicians with a more positive perception of SDM had a small but statistically significant effect on prescribing fewer antibiotics to patients with URTIs. However, no statistically significant difference was found for physicians’ prescribing of antibiotics and perception of SDM in female and male groups (OR = 0.92, *p* = 0.277).

As for the association between covariates and antibiotic prescribing behavior, female physicians (OR = 0.71, *p* = 0.007) with higher educational levels (bachelor’s degree and above, OR = 0.71, *p* = 0.024) were statistically found to be associated with the prescribing of fewer antibiotics. Compared with general practice, physicians in internal medicine and other specialties prescribed far fewer antibiotics (OR < 1, *p* < 0.001). Moreover, the setting of primary care facilities (THCs vs. CHCs) was one significant predictor of prescribing antibiotics, as THCs had a higher level of prescribing antibiotics (OR = 1.45, *p* = 0.012). The detailed information was presented in Table 3.

## 3. Discussion

In this study, applying the nine-item SDM instrument, physicians showed a strong positive perception of shared decision-making with patients with URTIs in primary care in Hubei province. The multilevel model showed physicians’ perception of SDM had small but statistically significant effects on prescribing fewer antibiotics in the patient group over 40 years old with URTIs in primary care.

Specifically, it is interesting to note that physicians with a higher perception of SDM prescribed fewer antibiotics in the patient group over 40 years old. This was partly in line with previous studies [10,23,24,25], where SDM interventions contributed to deliberation between physicians and patients and helped reduce antibiotic prescribing in situations in which the clinical guidelines or evidence does not clearly support using antibiotics or not (the study setting of URTIs in our study) [23,24]. The potential speculations as to why the relationship between perception of SDM and prescribing behavior worked in the patient group aged over 40 years were shown as follows.

Firstly, older patients whose health faces more complex conditions and who have a higher chance of comorbidity are more likely to regard the decision-making process, and they would more likely benefit from SDM with more discrete decisions as to whether or not to be treated with antibiotics [26].

Moreover, because older patients have a higher tendency to meet physicians’ expectations to be a good patient [18], physicians’ higher perception of SDM would facilitate greater participation of older adults in medical decisions, and thus made more discrete decisions about whether using antibiotics or not. However, as the data of socioeconomic characteristics as well as the health condition of the patients were not available in the outpatient prescription dataset, further research can be considered with the inclusion of these important indicators.

Secondly, there is sufficient evidence supporting the fact that older patients would more likely be involved in medical decisions [27,28,29]. Although some studies supported the idea that younger patients (below 65 years) preferred engaging in SDM, the surveys did not reveal the reasons behind the preference. Moreover, the perception of older patients’ unwillingness to participate in medical decisions conflicts with existing findings [18,30].

As for the lack of relationship between SDM and antibiotic prescribing in the patient group under 40 years, young patients’ low trust towards physicians would serve as one potential contributor. Of note, in China, it was found that young patients were less likely to trust physicians [31]. As a prerequisite for engaging with complex medical decisions, patients’ lower trust towards physicians would decrease both younger patients’ and physicians’ willingness to engage in SDM and deteriorate the physician–patient relationship. Further clarification and investigation are needed for exploring why no relationship was found in the patient group under 40 years old.

Except for the interaction effects between SDM perception and patient age, SDM also interacts with other intrinsic and extrinsic factors, such as the diagnosis uncertainty, complication of the decisions, or consultation time. For example, in the study setting of acute respiratory infections, several randomized controlled studies have demonstrated the positive effects of shared decision-making and C-reactive protein tests on fewer antibiotics prescribing in primary care [32,33]. The potential factors and interaction effects should be further explored in future research.

As for the associations between covariates and physicians’ prescribing of antibiotics, our findings mirrored previous research, in which higher educational level and female physicians were associated with less antibiotic prescribing in primary care [34,35]. General practice had substantially higher percentages of prescribing of antibiotics, which has been supported by several studies [36]. It was plausible that the overprescribing pattern of general practitioners, especially for acute respiratory infections, would influence physicians to be more likely to prescribe an antibiotic to patients with URTIs.

Moreover, the higher antibiotic prescribing rates of the rural township health centers in China have been documented in existing literature, particularly for respiratory tract infections. In a rural primary care setting, physicians are mostly non-degree-trained due to personnel shortage and thus were more likely to prescribe antibiotics irrationally [37].

### 3.1. Strengths and Limitations

This study attempted to match the prescribing data in practice and measurements of physicians’ questionnaire survey for being able to estimate the effects of physicians’ perception of SDM on antibiotic prescribing in practice. The nested structure of our data was adopted in this study, with prescription data at the patient level nested in each physician in primary care by conducting a multi-level logistic analysis.

Moreover, in contrast with the general judgment of perception of SDM, one strength of our study was that by extracting the electronic health records data, we focused on the context of the consultations with patients with URTIs. The choice of URTIs provides us the opportunity to evaluate the effects of shared decision-making on specific patients with specific consultations.

There were some limitations in this study. One limitation of this study is that it relied on physician self-reported results of the perception of SDM and thus may be at risk of social desirability bias [38]. The use of self-reported SDM-Q9 would result in an overestimation of the actual perception of SDM compared with analyses of objective audio or video recordings of physician–patient encounters, which would be considered in future studies. Secondly, as the study was conducted in primary care facilities in Hubei Province in China, replication of the studies in other different settings is desirable in order to examine the generalizability and robustness of our results.

### 3.2. Practice Implications

On the basis of the multilevel analysis results, there are important practical implications to emphasize the importance of SDM in primary care, especially under the context of URTI cases in the study setting. Since physicians’ perception of SDM was significantly associated with reduced antibiotic prescribing for patients with URTIs in the patient group over 40 years, this thus may be a promising focus of implementing SDM strategies in primary care.

Considering the primary care physicians’ challenges of inadequate education degrees, too short training, and insufficient practice, as well as the absence of humanities education in China [39,40], our findings call for sufficient education and training resources targeting physician–patient communication and shared decision-making in future continuing education, and thus help physicians and their patients to reduce overuse of antibiotic prescribing in primary care.

## 4. Materials and Methods

### 4.1. Study Setting

A cross-sectional study was conducted in Hubei, a province in central China with a population of 59.17 million, with social and economic development ranking in the middle of all the regions. This study focused on primary care facilities in Hubei, which covers both urban community health centers (CHCs) and rural township health centers (THCs). There were 1161 CHCs and 1139 THCs, serving 43.53 million outpatients in Hubei province in 2018, according to the latest Hubei Statistics Report [41].

### 4.2. Sampling

A multi-stage cluster sampling was applied in this study. A total of 102 primary care facilities were investigated from the provincial capital Wuhan and a random selection of other 4 prefecture-level cities in Hubei province. The primary care facilities were surveyed from December 2019 to January 2020. Details about the sampling approach have been published elsewhere [42]. The study was approved by the Ethics Committee of Tongji Medical College, Huazhong University of Science and Technology (IORG: IORG 0003571).

### 4.3. Pilot Study

One pilot study was conducted evaluating the readability and reliability of the 9-item shared decision-making questionnaire (SDM-Q9) (see details in Section 4.4, physicians’ questionnaire survey). The questionnaire was tested using a sample of 24 primary physicians in community health centers in Wuhan, Hubei Province. The respondents were asked to complete the questionnaire and provide oral feedback regarding each item’s readability. The initial Cronbach’s coefficient alpha of the SDM-Q9 was 0.78, and no items were deleted in the main study.

A two-stage data collection process was arranged: physicians’ questionnaire survey in Section 4.4 and prescription data extraction in Section 4.5.

### 4.4. Physicians’ Questionnaire Survey

The inclusion rules of the physicians were presented as follows: being able to prescribe antibiotics independently, and having a prescription of URTIs in adult patients ≥100 in the year of 2018. The physicians’ lists meeting the inclusion criteria were provided by the local health institutions according to the physicians’ profiles and their prescription data of 2018 before the survey.

The research teams were trained to contact the eligible physicians in order to conduct a face-to-face questionnaire survey. Before the survey, the introductory information “When a treatment decision needs to be made during a consultation with a patient presenting with acute respiratory infections (e.g., whether to use antibiotics or not), what are your perceptions of shared decision-making?” was provided for each participant. 

Physicians’ perception of SDM was measured by the 9-item shared decision-making questionnaire (SDM-Q9), which has been widely demonstrated with good feasibility, reliability, and validity [43,44]. The original SDM-Q9 was translated into Chinese and back-translated into English to ensure the accuracy of the translation, with a focus on ensuring translation of meaning and maintenance of context.

All responses were based on a 6-point Likert scale ranging from completely disagree (0) to completely agree (5). The overall score of the SDM-Q9 was calculated as described by Kriston et al., ranging from 0 to 100, where 0 indicates the lowest possible level of SDM and 100 indicates the highest possible level of SDM [43]. 

Physician demographic characteristics, such as gender, age, working years of practice, educational level, and receiving antibiotic training or not were also collected.

Specifically, it is worth noting the specific situation of medical education in China. Except for the bachelor’s program of medical college, junior medical college (leading to a vocational diploma) and secondary-level medical education (leading to a secondary vocational diploma) also exist due to the trade-off between quantity and quality of training physicians. Physicians without bachelor’s degrees are allowed to become assistant doctors, and they have the chance to obtain a full license after years of work experience and passing the National Practicing Doctor Examination [45].

### 4.5. Physicians’ Antibiotic Prescribing Data Extraction

Prescriptions issued by the questionnaire respondents were extracted from the 2018 outpatient prescription dataset. For each prescription with URTIs, whether antibiotics (identified on the basis of Anatomical Therapeutic Chemical Classification System, ATC, subgroup code J01) were prescribed or not was calculated as the dependent variable. The prescription data of the investigated physicians were extracted from the Hubei Province primary care outpatient prescriptions dataset from January 2018 to December 2018. The dataset is consistently collecting electronic clinical data of outpatient healthcare services in primary care facilities in Hubei Province covering, 2300 primary care facilities and 6.55 million citizens. 

The extraction rules are presented as follows: prescriptions with diagnosis judged by the physician to be an upper respiratory tract infection in adult patients (>18 years) were extracted.

### 4.6. Statistical Analysis

Descriptive statistics were used to calculate the average score of physicians’ SDM perception and antibiotic prescription for patients with URTIs. The reliability result showed SDM-Q9 yielded Cronbach’s coefficient alpha values of 0.82, which is greater than the minimum acceptable value of 0.70.

Multilevel logistic modeling was applied to examine the relationship between physicians’ perception of SDM and antibiotic prescribing. The data of measurement of SDM perception and physicians’ antibiotic prescribing for URTIs were matched for each physician. Whether antibiotics were prescribed for each prescription of URTIs was set as the dependent variable. Moreover, the interaction terms of age–SDM and gender–SDM were included in the model, as SDM behavior is known to be related to patient age and patient gender on the basis of existing literature [22]. Physicians’ demographic characteristics were also adjusted in the multilevel models. The statistical analyses were performed using STATA (version 12.0, Texas: StataCorp) and Mplus (version 6.0, Los Angeles, CA: Muthén and Muthén). A *p*-value < 0.05 was considered statistically significant.

## 5. Conclusions

In this study, a large sample of electronic prescription data nested in each physician from primary care facilities in Hubei, China, were investigated for evaluating the effects of physicians’ perception of SDM on prescribing behavior in primary care. Our findings demonstrated the relationship between a higher perception of SDM and less antibiotic prescribing behavior in the patient group over 40 years old with URTIs in primary care.

## Figures and Tables

**Table 1 antibiotics-09-00876-t001:** Demographics and antibiotic prescribing behavior of the participants.

Characteristics	*N* (%)/Median (IQR ^†^)	AB Rate ^‡^	*p*-Value
**Gender**			0.180
Male	170 (64.39)	0.66	
Female	91 (35.61)	0.53	
**Age (years)**	45 (38~50)	-	-
**Setting**			<0.001 *
Community health center (CHC)	94 (35.61)	0.52	
Township health center (THC)	170 (64.39)	0.67	
**Educational level**			<0.001 *
Secondary vocational education	45 (17.05)	0.69	
Vocational education	102 (38.64)	0.66	
Bachelor and above	117 (44.32)	0.55	
**Specialties**			0.037 *
General practice	177 (67.05)	0.64	
Internal medicine	57 (21.59)	0.55	
Others ^§^	30 (11.36)	0.60	
**Years of clinical practice (years)**	17 (9–25)	-	-
**Training regarding antibiotics last year**			0.447
Yes	215 (81.44)	0.61	
No/do not know	49 (18.56)	0.59	

Notes: ^†^ IQR, interquartile ranges (25th to 75th percentile); ^‡^ AB rate, antibiotic prescribing rate, calculated by the percentage of prescriptions containing antibiotics for each physician. ^§^ Other specialties included surgery and gynecology; they were combined due to the small sample size. * *p* < 0.05.

**Table 2 antibiotics-09-00876-t002:** The results of the shared decision-making (SDM) questionnaire (the SDM-Q9) of the participants (2019–2020).

Statements	Mean (SD)
I make clear to my patients that a decision needs to be made in a consultation.	3.95 (0.90)
I want to know exactly how my patients want to be involved in making the decision.	3.38 (1.21)
I tell my patients that there are different options for treating medical decisions.	3.98 (0.69)
I precisely explain the advantages and disadvantages of the treatment options of my patients.	4.15 (0.71)
I help my patients to understand all the information.	3.98 (0.74)
I ask my patients which treatment option they prefer.	3.49 (1.11)
I thoroughly weighted the different treatment options with my patients.	3.37 (1.21)
I select a treatment option together with my patients.	3.53 (1.06)
I reach an agreement with my patients on how to proceed.	3.97 (0.85)
Overall SDM Q-9 score ^†^	74.98 (12.08)

Note: ^†^ The overall SDM score for each respondent was calculated as described by Kriston et al., ranging from 0 to 100.

**Table 3 antibiotics-09-00876-t003:** Multilevel logistic regression of the relationship between physicians’ perception of SDM and prescribing behavior.

Independent Variables	OR ^†^	95% CI ^‡^	*p*-Value
SDM in reference group (reference group: age < 40 years, male)	1.01	0.999–1.017	0.070
Age (40–65 years)—SDM	0.99	0.988–0.999	0.021 *
Age (≥65 years)—SDM	0.98	0.982–0.996	0.002 *
Gender (female)—SDM	0.92	0.780–1.073	0.277
Covariates (physicians’ demographics)			
Physician’s age	0.99	0.984–1.015	0.949
Physician’s gender (reference group = male)	0.71	0.563–0.907	0.007 *
Years of practice	1.01	0.993–1.023	0.277
Education level (reference group = secondary vocational education)			
Vocational degree	0.97	0.737–1.293	0.868
Bachelor’s degree and above	0.71	0.524–0.955	0.024 *
Receiving antibiotics training	0.90	0.678–1.196	0.471
Specialties (reference group: general practice)			
Internal medicine	0.64	0.529–0.805	<0.001 *
Others	0.74	0.560–0.968	0.028 *
Facility setting (reference group: community health centers)	1.45	1.083–1.936	0.012 *

^†^ OR, odds ratio, higher showed a higher probability of antibiotic prescribing; ^‡^ CI, confidence interval; * *p* < 0.05. Abbreviations: SDM, shared decision-making.

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
