# Peer review of "Association between Physicians’ Perception of Shared Decision Making with Antibiotic Prescribing Behavior in Primary Care in Hubei, China: A Cross-Sectional Study"

_antibiotics, 2020, doi:10.3390/antibiotics9120876_

Round 1
Reviewer 1 Report
General:
in many places, spaces are missing before the brackets of references.
The paper should be checked for grammar mistakes.
L33: Please consider including the following reference in the paper:
Antibiotics 2019, 8(3), 129; https://doi.org/10.3390/antibiotics8030129
L35: Please consider including the following reference in the paper:
Antibiotics 2020, 9(9), 597; https://doi.org/10.3390/antibiotics9090597
L40-43: please rephrase this sentence or make 2 sentences. First, clearly define what SDM is.
L49: prefer more conservative treatment options
L59: a limited number of studies
L68: 121,481
Results:
The authors should make an effort to present their results in a more visually appealing way
Discussion:
L121: studies
L129: because older patients, to be a good patient
L136: some studies
Strenghts and limitations section: appropriate
Methods:
The “Physcians’ questionnaire survey” section should be a separate subsection in the MS.
References: they don’t adhere to the journal guidelines
Author Response
Dear Reviewers
Thank you for your letter as well as the comments concerning our manuscript entitled “Association Between Physicians’ Perception of Shared Decision Making with Antibiotic Prescribing Behavior in Primary Care in Hubei, China: A Cross-Sectional Study”. Those comments are all valuable and very helpful for revising and improving our paper. The comments have been studied very carefully and point-to-point revisions were made.
We appreciate for Editors and Reviewers’ warm work earnestly, and hope that the revisions will meet with approval. Revised portions are marked with Track Changes in the main document. The details of responses to the reviewer’s comments are as flowing in this word file:
Once again, thank you very much for your comments and suggestions.
Reviewer reports:
Reviewer 1
General: 1. in many places, spaces are missing before the brackets of references. The paper should be checked for grammar mistakes.
Answer: Thanks very much for your reminding and we checked the grammar mistakes throughout the manuscript and checked the missing spaces before references. The revisions were marked with Track Changes in the main document.
- L33: Please consider including the following reference in the paper:
Antibiotics 2019, 8(3), 129; https://doi.org/10.3390/antibiotics8030129
L35: Please consider including the following reference in the paper:
Antibiotics 2020, 9(9), 597; https://doi.org/10.3390/antibiotics9090597
Answer: Thanks very much for your comments and we have added the mentioned references as the Reviewer suggested.
The specific revisions were shown in the Reference section: Ref 2 and Ref 6, Page 9, Line 296, 303~305.
- L40-43: please rephrase this sentence or make 2 sentences. First, clearly define what SDM is.
Answer: Thanks very much for your comments and we are sorry for the confusion. We rephrased the sentences as the Reviewer suggested.
The specific revisions were shown as follows:
Introduction, Page 1-2, Line 40~42:
Shared decision making (SDM) is an approach where a physician and the patient make a decision together based on the best available evidence considering the patient’s values and preferences [9]. There is increasing evidence showed that recognized as one effective strategy for reducing the overuse of antibiotics in primary care [10-13].
- L49: prefer more conservative treatment options
Answer: Thanks very much for your comments and we revised the sentence as the Reviewer suggested.
The specific revisions were shown as follows:
Introduction, Page 2, Line 47~49:
With full consideration of the trade-off between the benefits and harms of antibiotics, most patients prefer tend to make more conservative treatment options [14].
- L59: a limited number of studies
Answer: Thanks very much for your comments and we revised the sentence as the Reviewer suggested.
The specific revisions were shown as follows:
Introduction, Page 2, Line 66~68:
however, a limited number of studies explored the relationship between SDM and fewer antibiotics with a focus on the general judgment of SDM, rather than for specific patients in specific consultations [11].
- L68: 121,481
Answer: Thanks very much for your comments and we revised the expression as the Reviewer suggested.
The specific revision is presented in Results, Page 2, Line 76~77.
In sum, the dataset of 121,481 prescription data nested in 264 primary care physicians was matched with completed SDM-Q9 measurements for each physician.
- Results:
The authors should make an effort to present their results in a more visually appealing way.
Answer: Thanks very much for your comments and we have tried to revise the format of the Tables in a more visually appealing way. The Tables and relevant description of the results were presented more concisely.
The specific revisions were shown in the main document: Page 3, Table 1&Table 2; Page 5, Table 3.
- Discussion:
- L121: studies
- L129: because older patients, to be a good patient
- L136: some studies
Answer: Thanks very much for your comments and we have revised the phases as the Reviewer suggested.
The specific revisions were presented as follows.
- Discussion, Page 6, Line 124~125:
This was partly in line with previous studies [10, 23-25].
- Discussion, Page 6, Line 133~136:
Besides, because older patients had a higher tendency to meet physicians’ expectations to be a good patient [18], physicians’ higher perception of SDM would facilitate greater participation of older adults in medical decisions, and thus made more discrete decisions about whether using antibiotics or not.
- Discussion, Page 6, Line 140~142:
Although some studies supported that younger patients (below 65 years) preferred engaging in SDM, the surveys, however, do not reveal the reasons behind the preference.
- Methods:
The “Physicians’ questionnaire survey” section should be a separate subsection in the MS.
Answer: Thanks very much for your comments and the “Physicians’ questionnaire survey” section and “Physicians’ antibiotic prescribing data extraction” were listed as two separate subsections in the Methods.
The specific revisions were presented in Materials and Methods.
Materials and Methods, Page 8, Line 215~217:
A two-stage data collection process was arranged: physicians’ questionnaire survey in subsection 4.4 and prescription data extraction in subsection 4.5.
4.4. Physicians’ questionnaire survey
4.5. Physicians’ antibiotic prescribing data extraction
- References: they don’t adhere to the journal guidelines
Answer: Thanks very much for your comment and we have downloaded the MDPI Endnote reference style from the official website, and we revised the reference style as the journal guidelines suggested.
To keep a neat presentation of the reference section, the application of the new MDPI style in the reference part was not marked with track changes in the Reference.
The specific revisions were presented in the Reference section, Page 9~12, Line 292~424.
Reviewer 2 Report
In this report, the authors conducted a cross-sectional study in Hubei province to measure physicians’ perception of Shared decision making (SDM) and establish the relationship between physicians’ perception of SDM and prescribing behavior in practice. This is an interesting work while several issues should be addressed before further consideration.
1. This study focus on the physicians’ prescription of antibiotic with patients with upper respiratory tract infections. Please revise the abstract to include the information of “patients with upper respiratory tract infections”.
2. Line 52: “A relatively small proportion of studies measured physicians’ perception of SDM in primary care, and most of them were conducted 10 years ago or involved relatively small groups of samples[18, 19].” Please carefully check the cited references and revise this statement. Actually, most of them were conducted within the recent 10 years. Besides, Ref 19 reported a large group of samples.
3. The introduction should be improved since it routinely introduces some background without real scientific depth. Specifically, the progress or positive results from previous publications in this field should be reasonably introduced rather than just emphasize their inadequacy.
4. It is not clear what types of physicians were involved in this study? For example, medical specialists, medical residents, etc.
Author Response
Dear Reviewers:
Thank you for your letter as well as the comments concerning our manuscript entitled “Association Between Physicians’ Perception of Shared Decision Making with Antibiotic Prescribing Behavior in Primary Care in Hubei, China: A Cross-Sectional Study”. Those comments are all valuable and very helpful for revising and improving our paper. The comments have been studied very carefully and point-to-point revisions were made.
We appreciate for Editors and Reviewers’ warm work earnestly, and hope that the revisions will meet with approval. Revised portions are marked with Track Changes in the main document. The details of responses to the reviewer’s comments are as flowing in this word file:
Once again, thank you very much for your comments and suggestions.
Reviewer 2
In this report, the authors conducted a cross-sectional study in Hubei province to measure physicians’ perception of Shared decision making (SDM) and establish the relationship between physicians’ perception of SDM and prescribing behavior in practice. This is an interesting work while several issues should be addressed before further consideration.
- This study focus on the physicians’ prescription of antibiotic with patients with upper respiratory tract infections. Please revise the abstract to include the information of “patients with upper respiratory tract infections”.
Answer:
Thanks very much for your comments and we are sorry for not clarifying the study context of the patient in the Abstract. We added the information of “patients with upper respiratory tract infections” in the Abstract as the Reviewer suggested.
Besides, to follow the author's instructions of “abstract should not exceed 200 words”, the conclusion part of the abstract was revised more concisely.
The specific revisions were shown as follows:
Abstract, Page 1, Line 13~15, Line 25~27:
Our study aims to measure physicians’ perception of SDM and establish the relationship between physicians’ perception of SDM and prescribing behavior in patients with upper respiratory tract infections.
There may be a promising focus of implementing SDM strategies targeting physician-patient communication in primary care.
- Line 52: “A relatively small proportion of studies measured physicians’ perception of SDM in primary care, and most of them were conducted 10 years ago or involved relatively small groups of samples[18, 19].” Please carefully check the cited references and revise this statement. Actually, most of them were conducted within the recent 10 years. Besides, Ref 19 reported a large group of samples.
Answer:
Thanks very much for your comments and we are sorry for the inaccurate statements. We have carefully checked the cited reference and revised the statement as the Reviewer suggested.
In the systematic review of Ref 18, around one-third of the included studies (14/43) considered SDM within the context of primary care. And after reviewing the studies conducted in the study setting of primary care, most of the studies only involve a sample size in primary care.
Also, although Ref 19 measured a large sample of physicians’ perception of shared decision making, the research was conducted in a large Dutch general teaching hospital, rather than in primary care facilities. To focus on the study setting of primary care, the original Ref 19 was removed in this study.
The specific revisions were presented as follows:
Introduction, Page 2, Line 56~58:
A relatively small proportion of studies measured physicians’ perception of SDM in primary care, and most of them involved relatively small groups of samples [20].
- The introduction should be improved since it routinely introduces some background without real scientific depth. Specifically, the progress or positive results from previous publications in this field should be reasonably introduced rather than just emphasize their inadequacy.
Answer:
Thanks very much for your comments. The progress and positive results concerning “physicians’ perception of shared decision making” and “the association between physicians’ perception of SDM and antibiotic prescribing” was supplemented in the background as the Reviewer suggested.
The specific revisions were presented as follows:
Introduction, Page 2, Line 50~54:
However, SDM does not occur consistently in clinical encounters [15-17]. Physicians’ perception and attitude is the key challenge of the implementation of SDM [18]. Many physicians feel they already involved the patient in treatment decisions, or physicians perceived themselves as “decision-makers” for patients[18]. To increase congruence between SDM theory and clinical practice, a thorough understanding of physicians’ perception of SDM is in high demand.
Introduction, Page 2, Line 55~59:
Existing researches revealed most studies of SDM were conducted in academic centers or public hospitals [19]. A relatively small proportion of studies measured physicians’ perception of SDM in primary care, and most of them involved relatively small groups of samples [20]. The physicians showed a preference towards SDM, although the level of supporting SDM depends on the clinical scenario, patient characteristics, and available treatment options [20].
Introduction, Page 2, Line 60~64:
In China, few researchers tried to introduce SDM intervention in a small sample of physicians in teaching hospitals and measured their preference of SDM, and most participants reported a positive preference towards SDM and decision aids [21, 22]. However, there is still a lack of understanding of how physicians perceived decision responsibilities and shared decision making in medical encounters in primary care.
Introduction, Page 2, Line 65~68:
On the other hand, interventions aiming at promoting SDM have been shown to reduce antibiotic prescription in primary care, however, a limited number of studies explored the relationship between SDM and fewer antibiotics with a focus on the general judgment of SDM, rather than for specific patients in specific consultations [11].
- It is not clear what types of physicians were involved in this study? For example, medical specialists, medical residents, etc.
Answer:
Thanks very much for your comments and we are sorry that we cannot fully sure that we correctly understand the meaning of “type of physicians”.
We tried to understand the type of physicians as the Specialties of the participants in primary care. In the primary care facilities, the majority of the participants were general practitioners. The information on Specialties was provided in the Results Section and Table 1. And the Specialties of the participants was adjusted in the multi-level analysis.
The information was presented in Results and Table 1(Page 3, Table 1).
Results, Page 2, Line 80~81:
Around two-thirds of the participants were from general practice (67.1%), and they had an average of 17 years of working practice.
Reviewer 3 Report
The present study aims to assess physicians’ perception of Shared decision-making (SDM) with patients with URTIs in primary care in China; and the also aims to explore the association between physicians’ perception of SDM and antibiotic prescribing in practice. The manuscript is prepared well and can be accepted as it is.
Author Response
Dear Reviewers:
Thank you for your letter as well as the comments concerning our manuscript entitled “Association Between Physicians’ Perception of Shared Decision Making with Antibiotic Prescribing Behavior in Primary Care in Hubei, China: A Cross-Sectional Study”. Those comments are all valuable and very helpful for revising and improving our paper. The comments have been studied very carefully and point-to-point revisions were made.
We appreciate for Editors and Reviewers’ warm work earnestly, and hope that the revisions will meet with approval. Revised portions are marked with Track Changes in the main document. The details of responses to the reviewer’s comments are as flowing in this word file:
Once again, thank you very much for your comments and suggestions.
The present study aims to assess physicians’ perception of Shared decision-making (SDM) with patients with URTIs in primary care in China; and the also aims to explore the association between physicians’ perception of SDM and antibiotic prescribing in practice. The manuscript is prepared well and can be accepted as it is.
Answer: Thanks very much for your comments and support. The minor revisions were marked with Track Changes in the main document.